# Angle-Insensitive Ultrathin Broadband Visible Absorber Based on Dielectric–Semiconductor–Lossy Metal Film Stacks

**DOI:** 10.3390/nano13192726

**Published:** 2023-10-08

**Authors:** Yuanchen Ma, Junhao Hu, Wenfeng Li, Zhengmei Yang

**Affiliations:** 1State Key Laboratory of Information Photonics and Optical Communications, Beijing University of Posts and Telecommunications, Beijing 100876, China; mayuanchen@bupt.edu.cn (Y.M.); hjh170211@bupt.edu.cn (J.H.); wfli@bupt.edu.cn (W.L.); 2School of Artificial Intelligence, Beijing University of Posts and Telecommunications, Beijing 100876, China; 3School of Science, Beijing University of Posts and Telecommunications, Beijing 100876, China

**Keywords:** broadband absorber, planar thin-film structure, ultrathin absorbing semiconductor, Fabry–Pérot cavity, interference effect

## Abstract

Ultrathin broadband absorbers with high efficiency, wide angular tolerance, and low fabrication cost are in demand for various applications. Here, we present an angle-insensitive ultrathin (<150 nm) broadband absorber with an average 96.88% (experiment) absorptivity in the whole visible range by utilizing a simple dielectric–semiconductor–lossy metal triple-layer film structure. The excellent broadband absorption performance of the device results from the combined action of the enhanced absorptions in the semiconductor and lossy metal layers exploiting strong interference effects and can be maintained over a wide viewing angle up to ±60°. Benefiting from the lossy metal providing additional absorption, our design reduces the requirement for the semiconductor’s material dispersion and has great flexibility in the material selection of the metal layer. Additionally, the lithography-free nature of the proposed broadband visible absorber provides a high-throughput fabrication convenience, thus holding great potential for its large-area applications in various fields.

## 1. Introduction

Broadband perfect absorbers, possessing near-unity absorption for the incident beam over a certain waveband along with the entirely suppressed reflection and transmission, have drawn substantial attention due to their great promise in many applications such as photovoltaic solar cells [1], thermal imaging [2], thermal emitters [3], and photodetectors [4]. Benefiting from the rapid development of nanofabrication technologies, broadband absorbers based on metamaterials and metasurfaces have been widely investigated due to their unique advantages of high absorptivity, ultrathin thickness, easy integration, and flexible working wavelength scalability compared to conventional heavy and bulky semiconductor absorbers [5]. In the past decade, various subwavelength micro/nanostructure systems such as tapered hyperbolic arrays [6], metal–insulator–metal (MIM) [7,8], nanogratings [9], nanoholes [10], nanowires [11], and nanocylinders [12] have been designed to realize broadband absorption properties across a wide frequency region from microwave to optical (even ultraviolet) bands by utilizing the coupling of multiple resonances to achieve a perfect impedance match between the air and the device [13,14,15,16]. However, the fabrications of the aforementioned broadband absorbers require multiple complicated and expensive nano-patterning processing steps such as e-beam lithography and reactive ion etching, greatly hindering their practical applications in a large area and low cost. Therefore, facilitating broadband perfect absorption in lithography-free planar thin-film structures is highly desirable and can be easily scaled up by using simple thin-film deposition technologies [17].

To date, various multilayer planar designs have been reported to achieve highly efficient absorption in a broad wavelength range [18]. A widely used configuration is a Fabry–Pérot (F-P) resonator consisting of dielectric–metal–dielectric–metal (DMDM) film stacks, in which the metal materials are generally selected from the metals with high losses, such as nickel (Ni), titanium (Ti), chromium (Cr), tungsten (W), and iron (Fe) [19,20,21,22,23]. The thickness of the top lossy metal layer is in the order of several nanometers, allowing the multiple round-trip reflections of the incident light within the structure and finally leading to sufficient light absorptions from the visible to the near-infrared (NIR) region by forming a low quality factor F-P cavity [24]. In addition, other strategies have also emerged to achieve broadband absorption performances by utilizing lossy metals [25,26,27]. For example, Yang et al. proposed an ultra-broadband absorber by continuously depositing ultrathin semiconductor (germanium (Ge), amorphous silicon (a-Si)), dielectric (titanium dioxide (TiO_2_), and magnesium fluoride (MgF_2_)) layers on a lossy Cr substrate [28]. The anti-reflection (AR) effects are introduced by the MgF_2_/TiO_2_/a-Si/Ge multilayers featuring the graded refractive index profile, resulting in a high absorption efficiency of 98% averaged from 400 to 2000 nm (even 20 μm) [29]. However, the structures of these absorbers involve plenty of layers (at least four-layer stack films), which makes the fabricating error and complexity, as well as cost, significantly increase. Recently, it was found that strong interference effects could be excited in an ultrathin absorbing semiconductor film coated on a metallic substrate, leading to strong absorption with a broad bandwidth at the resonant wavelength [30]. Although many broadband absorber designs have been proposed by utilizing ultrathin semiconductor films, the near-unity absorption characteristic cannot cover the whole visible range [2,31,32]. Therefore, how to achieve more efficient absorption over a wider spectral range with simpler planar thin-film structures still remains a significant challenge.

In this work, we propose and experimentally demonstrate a wide-angle ultrathin broadband visible absorber based on a triple-layer planar thin-film structure, which is comprised of a top dielectric layer, an ultrathin absorbing semiconductor film, and a lossy metallic substrate. Unlike a typical semiconductor–dielectric–metal mirror structure in which an unusual material dispersion is required for the semiconductor, our design is easier to implement and has great flexibility in the material selection of the metal layer. The fabricated device exhibits a nearly perfect absorption as high as 98.68% with an average absorption efficiency of 96.88% from 400 nm to 800 nm, which directly results from the combined action of the enhanced absorptions in the semiconductor and lossy metal layers by exploiting the excited F-P absorption and AR resonances. Additionally, due to the high refractive indices of the used materials and the ultrathin thickness (<150 nm) of the whole structure, the excellent broadband absorption performance of the device exhibits a great angular tolerance up to ±60°. The proposed broadband visible absorber can be realized using simple thin-film deposition methods, thereby opening up a new opportunity for its large-area and low-cost application in various areas, such as photovoltaic solar cells, imaging, and detections.

## 2. Methods

### 2.1. Device Fabrication

The proposed ultrathin broadband visible absorber based on dielectric–semiconductor–lossy metal triple-layer films was fabricated on a clean silicon substrate by e-beam evaporation (DE400, DE Technology Inc., King of Prussia, PA, USA) and atom layer deposition (ALD, R-200Adv, PICOSUN, Dresden, Germany). Ni and Ge films were deposited at room temperature with the rate of 2 Å/s and 0.5 Å/s, respectively, when the vacuum of the e-beam evaporator chamber was lower than 9 × 10^−8^ Torr. For TiO_2_ ALD deposition, the precursors used were titanium tetrachloride (TiCl_4_, pure 99.999%) as the metalorganic precursor and deionized water (DI water) as the oxidant. The deposition temperature was 80 °C where the chamber pressure was lower than 10 hPa. The carrier gas flows of TiCl_4_ and DI water were maintained at 120 sccm and 200 sccm, respectively. The average growth rate was measured to be 0.56 Å/cycle.

### 2.2. Simulation and Measurement

Optical simulations based on the transfer matrix method were carried out to calculate the absorption spectra of the designed broadband visible absorber at normal and oblique incidence, normalized electric field intensity distribution in the structure, optical admittance, and net phase shifts. The optical constants of all the materials were calibrated using a spectroscopic ellipsometer (RC2D, J. A. Woollam, Lincoln, NE, USA) and utilized in the simulations. An ultraviolet–visible–near-infrared spectrophotometer (Lambda-950, PerkinElmer, Waltham, MA, USA) and a spectroscopic ellipsometer (M-2000DI, J. A. Woollam, USA) were used to measure the reflection spectra of the fabricated device at normal incidence and 45°~70° viewing angles, respectively. The absorption performances were then obtained from 1-*R* relation, where *R* is the reflectance due to the zero transmittance of the device.

## 3. Results and Discussion

Figure 1a depicts the design process of our proposed angle-invariant broadband absorber based on an ultrathin planar thin-film structure. It begins with a simple bilayer structure consisting of an ultrathin absorbing semiconductor film on top of a highly reflective metallic mirror, in which a broadband absorption could be easily achieved at the resonant wavelength due to the excited strong interference effect [33]. However, the absorption efficiency is generally not sufficient. For example, the maximum absorptivity of ~82.57% appears at 550 nm, and the average absorption remains above around 65% over the visible regime for a typical design of 10 nm Ge/100 nm silver (Ag), as presented in Figure 1b(i). To further enhance the optical absorption, one widely used design is the semiconductor–dielectric–metallic mirror structure, in which the top semiconductor film is required to exhibit an unusual type of material dispersion [2,32,34]. Although an effective medium design using two absorbing films [34] and controlling the semiconductor’s polycrystallinity [32] was proposed to match the ideal optical complex refractive index, the near-unity absorption band still cannot cover the entire visible range, and the fabrication costs were also increased. Another design is the dielectric–semiconductor–metallic mirror structure, in which the top dielectric film acts as an AR layer to further suppress the reflection [35]. As shown in Figure 1b(ii), the reflection is significantly suppressed from 400 nm to 800 nm, and the average absorption efficiency is increased to ~88% when adding a 38-nm-thick TiO_2_ AR layer on the top of 10 nm Ge/100 nm Ag. It is worth noting that the reflection is still high at long wavelengths because the absorption is mainly ascribed to the Ge layer, while the intrinsic loss of Ge is very low in this region. Therefore, the highly reflective metal mirror can be further replaced by a lossy metal substrate, forming a dielectric–semiconductor–lossy metal structure to provide an additional absorption layer. It can be seen from Figure 1b(iii) that the reflection is further suppressed, and near-unity absorption performances are achieved in a wide wavelength range from 400 to 800 nm when replacing the Ag substrate with a lossy Ni layer. Compared to the typical semiconductor–dielectric–metallic mirror structure, this design reduces the requirements for the material dispersion of the semiconductor film and makes it easier to achieve broadband absorptions.

Figure 2a illustrates the schematic view of our designed lithography-free and angle-insensitive ultrathin broadband visible absorber, which is a triple-layer thin-film structure consisting of a 38-nm-thick TiO_2_ AR layer, a 10-nm-thick Ge absorbing semiconductor layer, and a 100-nm-thick Ni layer on a silicon substrate from the top to the bottom. The lossy Ni film with optical thickness can prevent any light transmission and act as an additional absorber at the same time, greatly reducing the limitations for the unique material dispersion of the ideal semiconductor required in other studies. Figure 2b provides the optical constants of these three materials used in calculations. Figure 2c plots the calculated and measured absorption spectra of the proposed device under normal incidence, presenting a fairly good match with each other and strong absorption performances across the entire visible range with average efficiencies of ~96.66% and ~96.88%, respectively. The maximum absorptivity of ~98.68% appears at 662 nm in the experiment, and two absorption peaks of ~98.31% and ~97.96% appear at 462 nm and 670 nm in the simulation, respectively. The inset in Figure 2b shows a photograph of the fabricated sample under ambient illumination, exhibiting a totally black color appearance at normal incidence. In addition, it is evident that our design has great flexibility in the material selection of the bottom metal layer. As presented in Figure 2d, broadband absorption performances with high efficiencies can also be easily achieved when using other lossy metal substrates, including Cr, Ti, and W. The optimized structural parameters are 42 nm TiO_2_/10 nm Ge/100 nm Cr, 40 nm TiO_2_/10 nm Ge/100 nm Ti, and 42 nm TiO_2_/15 nm Ge/100 nm W, respectively.

In order to better understand the mechanism of the strong broadband absorption performance in this triple-layer thin-film structure, the total absorption spectrum is plotted together with the separate absorption profiles in each layer of the device to elucidate the function of each layer, as depicted in Figure 3a. Obviously, the Ni and Ge layers contribute equally to the nearly perfect absorption across the entire visible range, but the short-wavelength absorption is mainly ascribed to Ge, and the long-wavelength absorption is concentrated in Ni, which can be clearly understood by studying the wavelength-dependent electric field distribution inside the designed device. As shown in Figure 3b, a strong electric field can be observed in Ni and Ge at all the wavelengths, and the stronger field intensity enters the Ni layer as the incident wavelength increases, which is consistent with the gradually enhanced absorption in Ni. It is interesting to note that although the electric field inside Ge is stronger in the long-wavelength range, the Ge layer exhibits a relatively weak absorption due to its intrinsically low loss property in this region. To reveal the resonant locations, the net phase shifts, which include two reflection phases at both the top and bottom interfaces and the propagation phase accumulation within the layer, are calculated for the Ge and top TiO_2_ layers and plotted in Figure 3c. The resonances occur at the wavelengths where the net phase change is equal to a multiple of 2π. It is found that an F-P absorption resonance is excited at 735 nm inside the Ge layer (the red curve), directly leading to the enhanced electric field and strong absorption in Ni at long wavelengths. In addition, the AR resonance @470 nm is excited within the top TiO_2_ layer (the blue curve), resulting in an enhanced electric field and strong absorption in Ge at short wavelengths. The F-P absorption and AR resonances function together, leading to the strong broadband absorption characteristic of the proposed device and creating two absorption peaks at 462 nm and 670 nm, respectively. Figure 3d compares the simulated absorption spectra of the layered structures with and without the top 38-nm-thick TiO_2_ film. It is evident that the device without the TiO_2_ layer (the red curve) exhibits a relatively flat absorption with an average absorption efficiency of 67.3% due to a considerable amount of reflection from the large index contrast between Ge and air. However, light absorption in the entire visible range is significantly enhanced by the excited AR resonance in the device with the top TiO_2_ layer (the black curve).

On the other hand, the highly efficient broadband absorption performance of the proposed ultrathin TiO_2_/Ge/Ni triple-layer film stacks can be explained by plotting the optical admittance diagram of the structure as well. The optical admittance (i.e., the reciprocal of the impedance) is defined as $Y=\sqrt{\varepsilon /\mu }$, where ε and μ are the relative permittivity and permeability, respectively, and is numerically equal to the material’s refractive index due to μ≈1 at optical frequencies for typical materials [36]. The admittance of the entire structure starts from the point of the silicon substrate, and the succeeding loci are determined by both the film thickness and the optical properties of the following layers. Lossless dielectrics and perfect electric conductors result in a circular trajectory, whereas spiral loci result from the absorbing media, such as semiconductors or metals. The distance between the ending admittance point of the film stacks and the admittance of air (1, 0) provides a direct measure of the reflectance of the entire structure’s reflectivity at normal incidence by utilizing the following equation:$$R={\left(\frac{1-\left(x+iy\right)}{1+\left(x+iy\right)}\right)}^{2},\;$$
where x and y are the real and imaginary parts of the final admittance, respectively. Figure 4 plots the admittance locus of our proposed ultrathin broadband visible absorber at different wavelengths, including 400 nm, 500 nm, 600 nm, 700 nm, and 800 nm, where the admittances present a similar variation trend and the ending admittance positions are (1.37, 0.62), (1.34, −0.11), (1.33, −0.21), (1.09, −0.3), and (0.91, −0.58), respectively. Obviously, these admittance points are very close to that of air, resulting in very low reflections of ~8.61%, 2.28%, 2.82%, 2.33%, and 8.7%, respectively. As the transmitted light is completely prevented by the bottom thick metallic substrate, the significantly suppressed reflections in the whole visible range lead to the broadband absorption property of the proposed device.

Next, we examine the influence of the thickness change in the TiO_2_ and Ge layers on the absorption efficiency of the proposed broadband visible absorber. Figure 5a provides a simulated 2D contour plot of the optical absorption as a function of the wavelength and the top TiO_2_ AR layer thickness for the designed TiO_2_/Ge/Ni triple-layer film stacks at normal incidence. The AR resonance will get shifted as the TiO_2_ thickness varies due to the different propagation phase shifts in the TiO_2_, directly resulting in a significant change in the absorption performance. It can be seen that the broadband absorption covering the entire working area with an average efficiency above 90% can be maintained at the TiO_2_ thickness ranging from 30 to 60 nm. Figure 5b plots the absorption spectra of the layered structure with the TiO_2_ thicknesses of 30, 40, 50, and 60 nm, respectively. Obviously, the absorption intensity decreases in the whole visible range when reducing the TiO_2_ thickness from 40 nm to 30 nm as the AR resonant condition is not satisfied. In contrast, with increased TiO_2_ thickness (>40 nm), the long-wavelength absorption is still very high due to the strong F-P absorption resonance excited in the Ge, but the short-wavelength absorption gradually decreases since the AR resonance gradually moves toward the longer wavelength range. Similarly, taking from the absorption plotted as a function of the wavelength and the Ge layer thickness in Figure 5c, the TiO_2_ thickness should be limited within the range of 5 to 20 nm to ensure the average absorption efficiency beyond 90%. Figure 5d plots the specific absorption spectra with the Ge thickness of 5, 10, 15, and 20 nm, respectively. As the Ge thickness increases from 5 nm to 10 nm, the F-P absorption resonance is further away from the AR resonance, resulting in enhanced light absorption at both short and long wavelengths. When further increasing the Ge thickness (>10 nm), the absorption resonance moves further towards the long-wavelength range and finally outside the working area, leading to a gradually reduced long-wavelength absorption. From this investigation, it is found that the optimal thickness for TiO_2_ and Ge should be chosen as 38 nm and 10 nm, respectively, in order to achieve the maximum overall absorption.

Lastly, we also explored the dependence of our proposed broadband visible absorber on the incident angle by calculating and measuring the angle-resolved absorption spectra under unpolarized light illumination, as illustrated in Figure 6a and Figure 6b, respectively. Obviously, the experimental observations seem in good agreement with the prediction of the calculated results, both presenting the relatively flat dispersion curves and highly efficient broadband absorption characteristics within a wide range of oblique incident angles from 0° to 70°. Specifically, the nearly broadband perfect absorption feature can be maintained before the value of the incident angle up to 40°, while there is a slight decrease in overall absorption as the incident angle further increases. The average absorption efficiencies of both the simulated and measured results are still higher than 90% and 75% at the very large incident angles of 60° and 70°, respectively. No significant reduction in the optical absorption indicates that the proposed broadband visible absorber exhibits a high angular tolerance up to ±60°, which can be further validated by the photographic images of the fabricated devices taken at different observing directions, as presented in Figure 6c. The device displays a stable high-purity black color appearance with negligible reflection as the viewing angle increases up to 60°. Such angle-insensitive absorption behavior directly results from the high refractive indices of the employed materials (i.e., Ge and TiO_2_), which can reduce the refraction angle inside the film stack according to Snell’s law and lead to a very small resonance shift when increasing the incident angle [13]. In addition, the accumulated propagation phase shifts within the structure are almost insignificant since the total thickness of the Ge and TiO_2_ films is much thinner than the incident wavelength, which is also responsible for the angle-robust broadband absorption performance of our designed device.

## 4. Conclusions

In summary, we present a new design scheme for an angle-insensitive ultrathin broadband visible absorber based on a simple triple-layer film structure consisting of an ultrathin absorbing semiconductor sandwiched between a dielectric layer atop and a lossy metal beneath. The lossy metal can provide additional absorption, reducing the requirements for the ideal semiconductor’s material dispersion in other designs and expanding the range of material selection of the metal layer. Under the action of the F-P absorption and AR resonances, the absorptions in the semiconductor and lossy metal are significantly enhanced, jointly leading to the near-unity absorption across the entire visible range with an average absorptivity of 96.88% in the experiment, respectively. As all the utilized materials feature high refractive indexes, the strong absorption performance of our device presents a great angle-invariant characteristic up to ±60°. The fabrication of the proposed broadband visible absorber only involves simple thin-film deposition processes, thus enabling its mass production for practical applications in various areas.

## Figures and Tables

**Figure 1 nanomaterials-13-02726-f001:**
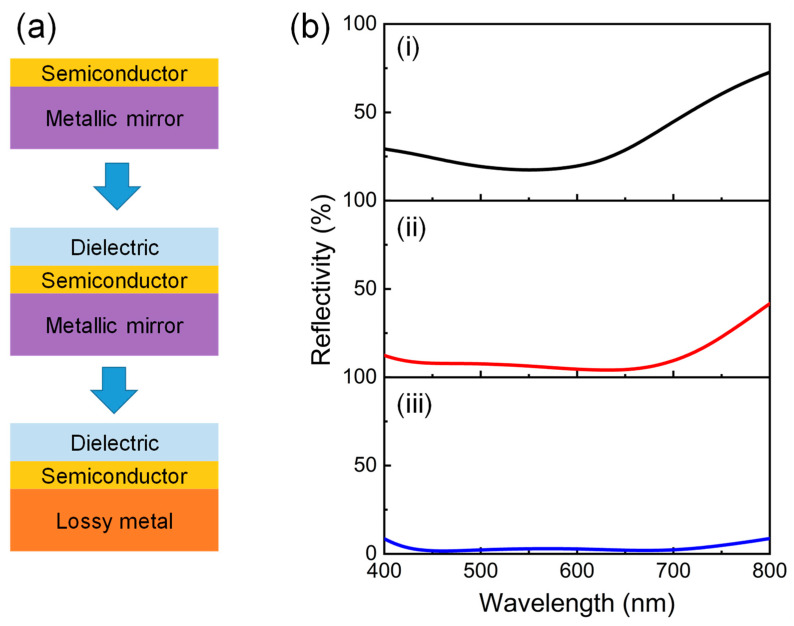
(**a**) Illustration of our broadband absorber design strategy. The configurations from top to bottom are an ultrathin semiconductor film coated on a highly reflective metallic mirror, a semiconductor film coated on a metallic mirror with an additional dielectric added, and a semiconductor film coated on a lossy metal substrate with an additional dielectric added. (**b**) The simulated reflection spectra of the typical broadband absorber designs in (**a**): (**i**) 10 nm Ge/100 nm Ag; (**ii**) 38 nm TiO_2_/10 nm Ge/100 nm Ag; and (**iii**) 38 nm TiO_2_/10 nm Ge/100 nm Ni.

**Figure 2 nanomaterials-13-02726-f002:**
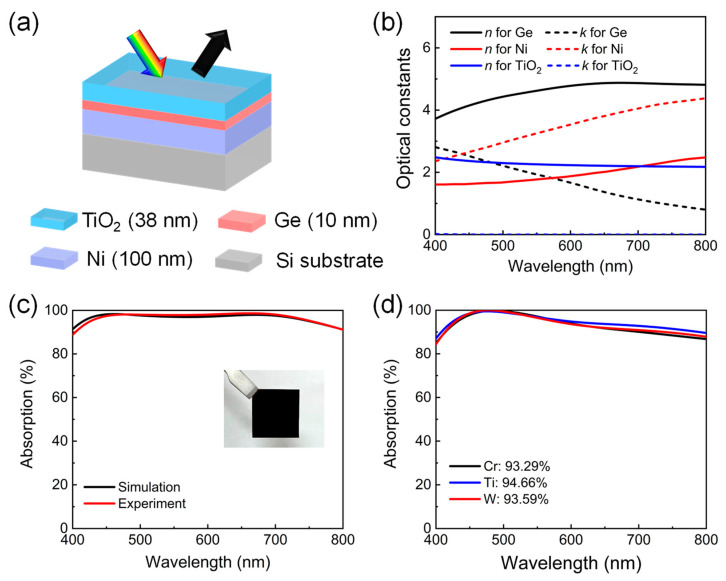
(**a**) The three-dimensional schematic illustration of our designed ultrathin broadband visible absorber. (**b**) The optical constants of lossy materials and dielectric materials used for the calculations. (**c**) The simulated and measured absorption spectra of the proposed device at normal incidence. The inset shows the optical image of a fabricated sample with dimensions of 2 cm × 2 cm. (**d**) The simulated absorption spectra of the optimized broadband visible absorber using different lossy metal substrates, including Cr, Ti, and W. Their corresponding average absorption efficiencies are also calculated.

**Figure 3 nanomaterials-13-02726-f003:**
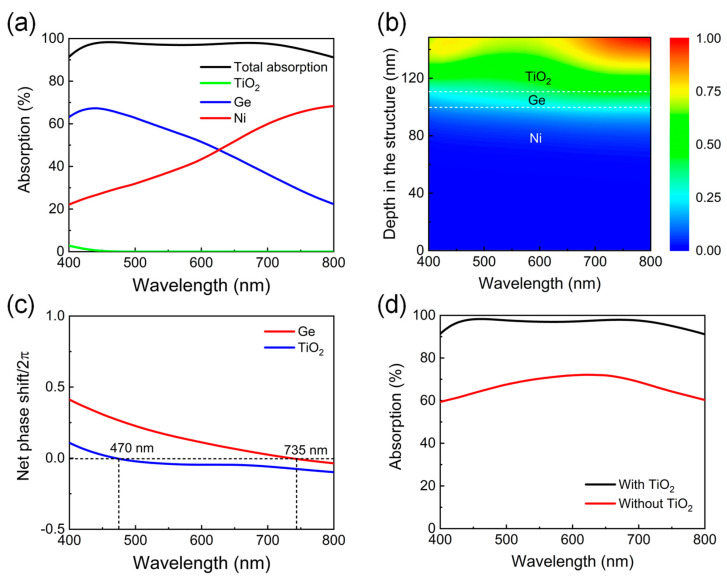
(**a**) Total absorption spectra of the proposed ultrathin broadband visible absorber and separate absorption in each layer. (**b**) Normalized electric field intensity profiles inside the whole structure of the device at all wavelengths. (**c**) Calculated net phase shifts divided by 2π within the top TiO_2_ and Ge layers, respectively. (**d**) Absorption spectra comparisons of the device with and without the top TiO_2_ AR layer.

**Figure 4 nanomaterials-13-02726-f004:**
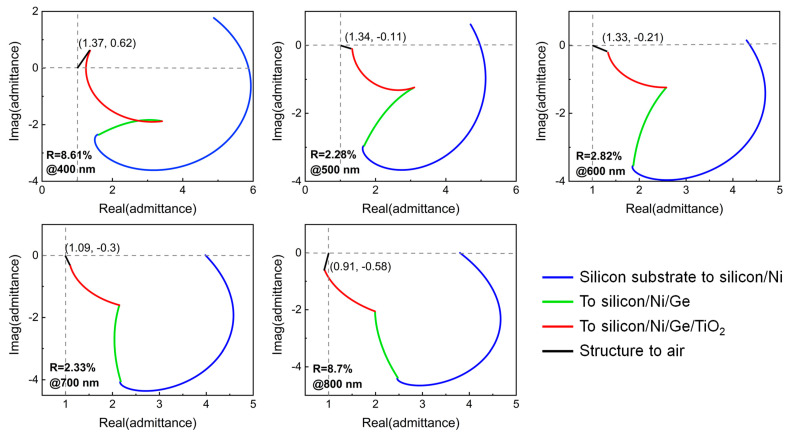
Optical admittance locus of the proposed broadband visible absorber at five discrete wavelengths in the visible range. The reflectance of the device at normal incidence is proportional to the length of the black line, which connects the final admittance point of the structure and the point of the incident medium (air).

**Figure 5 nanomaterials-13-02726-f005:**
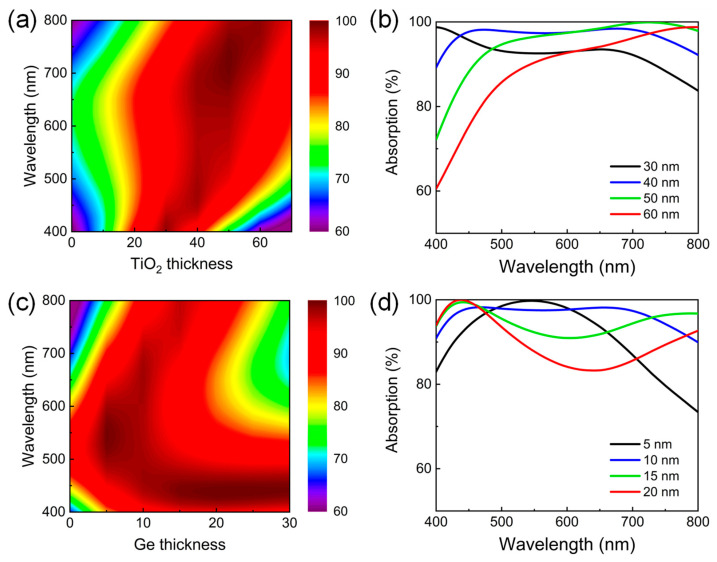
(**a**,**b**) The effect of the TiO_2_ layer thickness on the optical absorption performance of the designed TiO_2_/Ge/Ni triple-layer broadband visible absorber with the Ge thickness fixed at 10 nm. (**c**,**d**) The absorption variation of the device with the Ge layer thickness when maintaining the TiO_2_ thickness as 38 nm.

**Figure 6 nanomaterials-13-02726-f006:**
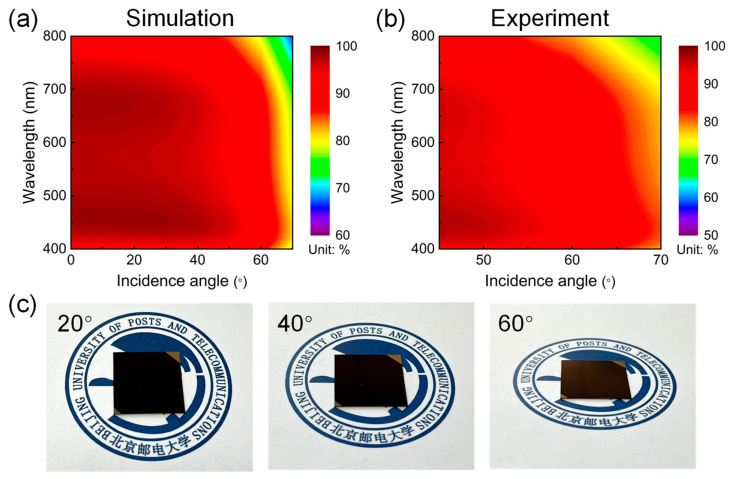
(**a**) The simulated and (**b**) measured angle-resolved absorption spectra of the proposed ultrathin broadband visible absorber under unpolarized light incidence up to 70°. (**c**) Photographic images of the fabricated broadband absorber device taken with indoor ambient light at the viewing angles of 20°, 40°, and 60°. The sample dimension is 2 cm × 2 cm.

## Data Availability

The data that support the findings of this study are available from the corresponding author upon reasonable request.

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
