# Peer review of "Angle-Insensitive Ultrathin Broadband Visible Absorber Based on Dielectric–Semiconductor–Lossy Metal Film Stacks"

_nanomaterials, 2023, doi:10.3390/nano13192726_

Round 1
Reviewer 1 Report
This manuscript provides simple multilayer structure for broad-range near-unity absorber structure. I recommend publishing this manuscript after some minor revisions.
1. The suggested structure consist of 3 layers, Insulator/semiconductor/lossy metal and the Introduction part describing disadvantage of multi-layer based absorber structure. In my sense, it is hard to support your motivation. Instead of including that part, I suggest to consider previous research exploiting 2D material, phase-changing materal, and so on.
2. Thickness of each layers in your device are provided in the manuscript, but there is not much description on the process. I would like ask to provide clear informaton wheter those values are nomimal thickness, or measurements are conducted with any specific method.
3. In section 2.2, It is described that optical constants were calibrated based on ellipsometry result. Please add your calibrated data used for calculation.
The quality of English writing is acceptable.
Author Response
Reviewer #1: This manuscript provides simple multilayer structure for broad-range near-unity absorber structure. I recommend publishing this manuscript after some minor revisions.
→ The authors thank the reviewer for this positive comment.
- The suggested structure consist of 3 layers, Insulator/semiconductor/lossy metal and the Introduction part describing disadvantage of multi-layer based absorber structure. In my sense, it is hard to support your motivation. Instead of including that part, I suggest to consider previous research exploiting 2D material, phase-changing materal, and so on.
→ We thank the reviewer for highlighting this point. Compared to the broadband absorber based on subwavelength nanostructure systems involving complicated and expensive nano-patterning fabrication technologies, multi-layer based absorber structures exhibit greater potential toward large-scale applications in various areas. However, a critical problem for multi-layer based broadband absorber is that plenty of layers are required to achieve relatively high absorption efficiency, and the wider the bandwidth, the more layers there are. It will lead to a new requirement for multiple deposition steps and increase the complexity of the manufacturing processes. Therefore, it is highly desirable to design broadband absorber with simpler thin-film structures. Besides, the near-unity absorption characteristic cannot cover the whole visible range for the current broadband absorbers utilizing the simplest triple-layer structures. So our motivation is to design a broadband visible absorber featuring high efficiency based on a triple-layer thin-film structures. In the revised manuscript, we add two references related to broadband absorber exploiting 2D material and phase-changing material, they are:
[15] Hashemi, M.; Ansari, N.; Vazayefi, M. MoS2-based absorbers with whole visible spectrum coverage and high efficiency. Sci. Rep. 2022, 12, 6313.
[16] Cao, T.; Wei, C.W.; Simpson, R.E.; Zhang, L.; Cryan, M.J. Broadband polarization-independent perfect absorber using a phase-change metamaterial at visible frequencies. Sci. Rep. 2014, 4, 3955.
- Thickness of each layers in your device are provided in the manuscript, but there is not much description on the process. I would like ask to provide clear information wheter those values are nomimal thickness, or measurements are conducted with any specific method.
→ We thank the reviewer for highlighting this point. In our process, each layer with a target thickness of 15 nm is deposited on a substrate separately, the real thickness and optical constant of the film are calibrated by an ellipsometer. Then the thickness of each layer in our device to realize broadband visible absorption is optimized based on the calibrated optical constants. Lastly, the device is fabricated according to the optimized structural parameters. The calculated absorption spectrum of the device under normal incidence presents a fairly good match with the measured one (Figure 2c), showing that the fabrication error (i.e. thickness) is very small.
- In section 2.2, It is described that optical constants were calibrated based on ellipsometry result. Please add your calibrated data used for calculation.
→ We thank the reviewer for the valuable suggestion. The optical constants of lossy materials and dielectric materials that are used for the calculations have been added as Figure 2(b) in our manuscript (line 148-151).
Reviewer 2 Report
The authors report on the design, fabrication and experimental testing of an efficient absorber of visible light. They note that a thin Ge layer deposited on a relatively thick lossy metal film facilitates broadband light absorption which can be further enhanced by a transparent dielectric antireflection coating. Optimizing the thickness of the layers allowed achieving almost perfect (96.88% on average) absorption in a broad (400-800nm) visible range which appears to be stable for the angles of incidence up to 60 degrees from the normal. These functional parameters are indeed remarkably better than those of the previously reported analogs. The absorber design is also fairly simple as it consists of only three layers (dielectric-semiconductor-lossy metal) being also not very sensitive to the particular choice of the lossy metal.
The paper is very well written. The main conclusions are supported by the simulations and, importantly, by the experimental tests. All this convinces me that the paper is appropriate for Nanomaterials. However, I came across a few issues to be addressed before publication:
1. The absorption is declared to be evaluated by the known reflectance (line 109). How was it possible to evaluate the partial absorptions in different layers plotted in Fig. 3a)?
2. The authors provide a rather extensive review of various absorber designs but somehow avoid mentioning the very well known paper [M. Chirumamilla et al., Multilayer Tungsten-Alumina-Based Broadband Light Absorbers for High-Temperature Applications, Opt. Mater. Express 6, 2704 (2016)]. Note that the design proposed there is not much more sophisticated, while the absorption performance is comparable.
3. The authors use the term “AR resonance” which likely corresponds to “anti-reflection resonance”. I am unfamiliar with this optical phenomenon. Please explain, provide references or reformulate.
